# Anti-Cancer and Ototoxicity Characteristics of the Curcuminoids, CLEFMA and EF24, in Combination with Cisplatin

**DOI:** 10.3390/molecules24213889

**Published:** 2019-10-29

**Authors:** Jerry D. Monroe, Denis Hodzic, Matthew H. Millay, Blaine G. Patty, Michael E. Smith

**Affiliations:** Department of Biology, Western Kentucky University, 1906 College Heights Boulevard, #11080, Bowling Green, KY 42101-1080, USA; jerry.monroe@wku.edu (J.D.M.); denis.hodzic021@topper.wku.edu (D.H.); matthew.millay602@topper.wku.edu (M.H.M.); blaine.patty739@topper.wku.edu (B.G.P.)

**Keywords:** cancer, curcuminoid, cisplatin, zebrafish, reactive oxygen species, auditory evoked potential, apoptosis, cell migration

## Abstract

In this study, we investigated whether the curcuminoids, CLEFMA and EF24, improved cisplatin efficacy and reduced cisplatin ototoxicity. We used the lung cancer cell line, A549, to determine the effects of the curcuminoids and cisplatin on cell viability and several apoptotic signaling mechanisms. Cellular viability was measured using the MTT assay. A scratch assay was used to measure cell migration and fluorescent spectrophotometry to measure reactive oxygen species (ROS) production. Western blots and luminescence assays were used to measure the expression and activity of apoptosis-inducing factor (AIF), caspases-3/7, -8, -9, and -12, c-Jun N-terminal kinases (JNK), mitogen-activated protein kinase (MAPK), and proto-oncogene tyrosine-protein kinase (Src). A zebrafish model was used to evaluate auditory effects. Cisplatin, the curcuminoids, and their combinations had similar effects on cell viability (IC_50_ values: 2–16 μM) and AIF, caspase-12, JNK, MAPK, and Src expression, while caspase-3/7, -8, and -9 activity was unchanged or decreased. Cisplatin increased ROS yield (1.2-fold), and curcuminoid and combination treatments reduced ROS (0.75–0.85-fold). Combination treatments reduced A549 migration (0.51–0.53-fold). Both curcuminoids reduced auditory threshold shifts induced by cisplatin. In summary, cisplatin and the curcuminoids might cause cell death through AIF and caspase-12. The curcuminoids may potentiate cisplatin’s effect against A549 migration, but may counteract cisplatin’s effect to increase ROS production. The curcuminoids might also prevent cisplatin ototoxicity.

## 1. Introduction

The platinum-based chemotherapy compound *cis*-diamminedichloroplatinum(II) (cisplatin) (Figure 1) and the phytochemical curcumin can both reduce cancer cell viability and migration [1,2,3,4]. Cisplatin treatment damages DNA and can act through multiple apoptotic pathways by increasing toxic intracellular reactive oxygen species (ROS) to kill cancer cells [1,5]. Unfortunately, cisplatin’s ability to increase apoptosis and ROS production leads to side effects, including ototoxicity and hearing loss [6,7,8,9,10]. Curcumin acts against cancer by targeting a broad set of pathways, including the cell cycle, apoptotic mechanisms, microRNAs, the proteasome, Wnt/β-catenin, and NF-κB signaling, as well as several protein kinases [11]. Curcumin can also cause increased toxic intracellular ROS release in cancer cells, but is able to scavenge and decrease ROS levels and act as an otoprotectant in non-cancer cells [12,13,14]. In non-cancer cells, curcumin can directly react with and inactivate several ROS and can also upregulate antioxidant and cytoprotective proteins [12]. However, in cancer cells, curcumin can target pro-oxidant pathways incorporating the transcription factors STAT3 and Nrf-2, which can also modulate increased sensitivity to cisplatin treatment [13].

The generally distinct mechanistic action of cisplatin and curcumin in cancer cells suggests that combining them could increase their anticancer effect. However, curcumin exhibits limited bioavailability, which restricts its efficacy [2,15]. Synthetic curcumin derivatives (curcuminoids) have now been developed that have anti-cancer activity and improved bioavailability [16,17]. The curcuminoids 4-[3,5-bis[(2-chlorophenyl)methylene]-4-oxo-1-piperidinyl]-4-oxo-2-butenoic acid (CLEFMA) and (3*E*,5*E*)-3,5-bis[(2-fluorophenyl)methylene]-4-piperidinone (EF24) have superior solubility and possess antiproliferative activity [18,19]. CLEFMA causes death in lung cancer cells via ROS production and activation of multiple apoptotic effector molecules [19,20,21,22]. EF24 also kills cancer cells by signaling through multiple apoptotic mechanisms, but the role of ROS production in EF24-mediated cell death is uncertain [23,24,25]. Although cisplatin, CLEFMA, and EF24 affect similar components in apoptotic signaling pathways, dissimilarities between cisplatin and these curcuminoids have been identified, including their effect on glutathione signaling, which is implicated in cisplatin-resistance [1,19,24]. However, the effect of cisplatin in combination with CLEFMA or EF24 on cancer cell viability, cell migration, apoptotic signaling, or how these curcuminoids might affect cisplatin’s auditory side effects are currently not well understood.

## 2. Results

We utilized a series of biochemical assays along with a zebrafish model of hearing to evaluate whether CLEFMA and EF24 could promote the anti-cancer action of cisplatin and prevent the platinum compound’s auditory side effects. First, we used the MTT assay to ascertain the inhibitory concentration at 50% (IC_50_) values of cisplatin, CLEFMA, and EF24 alone and combination treatments in which either one of the curcuminoids was combined with cisplatin.

We found that cisplatin had a 24-h IC_50_ value of 10.91 µM ± 0.19 and a 48-h value of 7.49 µM ± 0.16 (Table 1). In previous work [26], we obtained a 72-h value for cisplatin of 9.79 µM ± 0.63 in the A549 cell line and decided to use 10 µM cisplatin for the remainder of the experiments. We also found that EF24 had a lower 24-h IC_50_ value, 1.74 ± 0.28, than CLEFMA, which had a value of 13.82 ± 0.18 (Table 1). The IC_50_ values for the curcuminoids during the second half of a 48-h time interval were found to be 2.47 ± 0.14 for EF24 and 16.05 ± 0.15 for CLEFMA, which we rounded to 2 and 15 µM, respectively, for subsequent experiments (Table 1). We also obtained IC_50_ values of 2.19 µM ± 0.17 for cisplatin together with CLEFMA and 2.94 µM ± 0.09 for cisplatin together with EF24. In addition, cellular viability plots for all of the treatments were produced (Appendix A). As a final step, we performed a Bliss independence analysis of the combination treatments compared to cisplatin or either curcuminoid alone and found that the combination treatments exhibited an additive and not synergistic effect (Appendix A).

Fluorescent spectrophotometry was used to measure ROS production in A549 cells treated with the different compounds and cisplatin–curcuminoid combinations. The positive control and cisplatin treatments caused significantly increased ROS yield compared to the negative control (positive, 1.17-fold; cisplatin, 1.20-fold; Figure 2). However, the curcuminoid and cisplatin with curcuminoid treatments had significantly reduced ROS levels compared to the negative control (CLEFMA, 0.81-fold; EF24, 0.85-fold; cisplatin–CLEFMA, 0.76-fold; cisplatin–EF24, 0.75-fold; Figure 2). Also, we found that the combination treatments had significantly less ROS yield than EF24 but not CLEFMA (Figure 2).

We then used a luminogenic assay to measure the activity of several caspases that can be modulated during cisplatin-mediated apoptosis. First, we measured the effect of our experimental preparations on caspase-8 and found that EF24, the combination treatments, and the positive control caused no significant change in activity compared to the negative control (EF24, 0.67–fold; cisplatin–EF24, 0.67-fold; cisplatin–CLEFMA, 0.65–fold; positive control, 1.16-fold; *p* > 0.05, Figure 3A). However, cisplatin and CLEFMA had significantly decreased activity (cisplatin, 0.58-fold; CLEFMA, 0.43-fold compared to control). Caspase-9 activity did not change for cisplatin with EF24 treatment (0.68-fold compared to control; *p* > 0.05) and significantly increased for the positive control (1.32-fold of control), but caspase-9 activity decreased for all other treatments (cisplatin, 0.59-fold; EF24, 0.68-fold; CLEFMA, 0.48-fold; cisplatin–CLEFMA, 0.75-fold; Figure 3B). We then measured the activity of caspase-3/7, which integrates caspases-8 and -9, and found that activity compared to the negative control did not change for the combination treatments (cisplatin–EF24, 0.90-fold; cisplatin–CLEFMA, 0.89-fold), while activity for the positive control significantly increased (1.43-fold compared to negative control) and activity decreased for the cisplatin, CLEFMA, and EF24 treatments (cisplatin, 0.70-fold; EF24, 0.71-fold; CLEFMA, 0.74-fold; Figure 3C).

We then investigated whether cisplatin, the curcuminoids, or cisplatin–curcuminoid combinations could affect cancer cells migration using a wound recovery assay (Figure 4). We found that cisplatin, CLEFMA, and EF24 treatment did not significantly prevent recovery into cleared areas compared to controls during a 24-h interval (Figure 5). However, when we measured the effect of combinations of cisplatin with CLEFMA and EF24 against wound recovery, we found that migration was significantly reduced compared to control (Figure 5). Our results showed the following percent wound recovery value profile: Control (25.0%), CLEFMA (22.8%; 0.91-fold of control), EF24 (21.9%; 0.88-fold of control), cisplatin (19.1%; 0.76-fold of control), cisplatin–CLEFMA (12.8%; 0.51-fold of control), and cisplatin–EF24 (13.3%; 0.53-fold of control). A dimethyl sulfoxide (DMSO) control was not significantly different than the media vehicle control (24.8%; 0.99-fold of control; *p* > 0.05).

Western blot analysis was used to determine the effect of cisplatin, the curcuminoids, and their combination on the expression of several proteins involved in cisplatin-mediated signaling. We found that all treatment categories exhibited expression of apoptosis-inducing factor (AIF) protein (Figure 6). Similarly, we also performed a blot analysis of the protein, caspase-12, and were able to detect this protein for all categories except for EF24 (Figure 6). We measured the expression of mitogen-activated protein kinase (MAPK) and phosphorylated MAPK (pMAPK) and found both forms in all treatment samples (Figure 6). We then investigated the effect of the compounds and combinations on the expression of c-Jun N-terminal kinases (JNK) and phosphorylated JNK (pJNK) and detected expression for all treatment categories of the non-phosphorylated but not the phosphorylated form (Figure 6). As a final analysis, we also characterized the effect of the test compounds and combinations on proto-oncogene tyrosine-protein kinase (Src) signaling. We found that all treatments were associated with Src expression but that the phosphorylated form (pSrc) was not detectable in the blots (Figure 6). We also found that throughout the blots, samples treated with EF24 had noticeably decreased protein expression compared to the other treatments. Also, cisplatin-treated pMAPK samples and cisplatin in combination with CLEFMA-treated Src samples had visibly reduced band intensity compared to the other treatment categories.

As a final step, we performed the auditory evoked potential technique in a zebrafish model of hearing to determine if treatment with cisplatin, the curcuminoids, and the cisplatin–curcuminoid combinations altered hearing. We found that fish injected with cisplatin exhibited threshold shifts at several frequencies compared to sodium chloride (NaCl) vehicle- or DMSO vehicle-injected fish (Figure 7A; Table 2 and Table 3; Appendix A). When a curcuminoid was injected and compared to its DMSO vehicle control, EF24 and CLEFMA had decreased threshold shifts and were significantly different from controls at fewer frequencies compared to those of cisplatin (Figure 7B; Table 3; Appendix A). We then investigated whether injecting a curcuminoid 24 h after an initial treatment of cisplatin would significantly alter hearing threshold shifts compared to cisplatin injection alone. Our results showed that the subsequent injection of both CLEFMA and EF24 caused threshold shifts to decrease compared to control (Figure 7C; Table 2 and Table 3; Appendix A). Also, a comparison of cisplatin-treated fish, with fish first injected with cisplatin and then treated with DMSO 24 h later, indicated that the cisplatin–DMSO combination produced an audiogram similar to that of cisplatin–CLEFMA (Figure 7C; Table 3; Appendix A). Further, when we compared mean temporary threshold shifts between cisplatin-, cisplatin–curcuminoid-, and cisplatin–DMSO-treated fish, we found that cisplatin’s mean threshold shifts were not significantly increased compared to cisplatin–EF24 treatment, but that cisplatin caused a significantly greater threshold shift than the cisplatin–CLEFMA and cisplatin–DMSO treatments (both *p* < 0.0001; Figure 7D).

## 3. Discussion

Combining cisplatin with anti-cancer compounds that act through different pathways could improve anti-cancer efficacy while potentially reducing the platinum compound’s side effects [27,28]. Both CLEFMA and EF24 have anti-cancer activity, and EF24 can potentiate the effect of cisplatin against cancer cells [19,20,29,30]. However, the mechanistic interaction of cisplatin with either CLEFMA or EF24 is not well-understood, and the effect of these curcuminoids on ototoxicity has not been studied. We first used the MTT assay to measure the effect of cisplatin, CLEFMA, EF24, and combinations of cisplatin with either curcuminoid on A549 cell culture viability. Our results showed that all three compounds reduced cellular viability in the order: EF24 < cisplatin < CLEFMA (Table 1). Our analysis of the combination of cisplatin and either EF24 or CLEFMA suggests that the curcuminoids are unable to potentiate the effect of cisplatin (Appendix A), and we note that combination treatment cell viability did not decrease below the IC_50_ value of EF24 (Table 1). However, the absence of synergistic effects between cisplatin and either curcuminoid against cellular viability does not rule out action through separate cell signaling pathways.

Cisplatin, curcuminoid, and combination treatments could affect A549 cancer cell viability by acting on multiple signal transduction mechanisms. Therefore, we used luminogenic and blot assays to investigate the activity and expression of several proteins that modulate cancer cell viability. Cisplatin and dietary phytochemical treatment can increase the expression of and translocation of AIF from the mitochondria to the nucleus, where AIF can then cause DNA fragmentation and cell death without activating caspase-based mechanisms [31,32]. Our caspase and blot assay results showed that caspase activity was either suppressed or unchanged and that AIF expression was detectable, which could suggest that cell death signaling in the A549 cells might not incorporate caspases, but could possibly utilize AIF signaling instead and be responsible for the reduction in cell viability that we observed (Table 1; Figure 3 and Figure 6). Alternatively, by acting as a NADH oxidase, AIF signaling could cause ROS stress, MAPKs, and JNKs, which are integrated into cancer apoptosis, proliferation, differentiation, and survival responses [33,34]. This interpretation may be supported by our MAPK blot analysis results, where we detected both MAPK and pMAPK in our samples (Figure 6). We were surprised to find that cisplatin, the curcuminoids, and the cisplatin–curcuminoid combinations caused decreased or unchanged caspase activity, despite our positive control inducing significant caspase-3/7 and -9 activity (Figure 3). This result could mean that our treatments initially caused suppression of cell viability, but that by the later 48-h time point, the treatments also began to activate cell survival mechanisms associated with MAPK. In cell line studies, after initial treatment with cisplatin and other drugs, cell survival and resistance mechanisms can be modulated via activation of MAPK signaling [35,36]. Therefore, our blot analysis could be showing that curcuminoid and cisplatin treatment causes suppression of cell viability, followed by a later phase of cell survival and resistance signaling demonstrated by the expression of MAPK and pMAPK (Figure 6).

The effect of cisplatin and the curcuminoids against A549 cell viability could also integrate caspase-12 and Src signaling that separately can act upstream of MAPK and JNK. Drugs that induce ROS stress on the endoplasmic reticulum in non-small cell lung cancer cells can activate caspase-12 signaling and downstream MAPK signaling [37]. Our blot data showed banding for MAPK, pMAPK and caspase-12 for all treatment categories, except for EF24 in the caspase-12 blot (Figure 6), suggesting potential caspase-12 signaling in the A549 cells. Activation of the MAPK pathway via caspase-12 could act in concert with MAPK signaling downstream of AIF to initially promote apoptosis and diminished cell viability followed by later activation of cell survival mechanisms. Activation of survival pathways by caspase-12 signaling is also consistent with the reduced or unchanged caspase-3/7, -8, or -9 activity that we observed (Figure 3). Src signaling could also act upstream of both MAPK and JNK to modulate apoptosis and survival mechanisms [38,39]. However, our blot analysis, which detected both Src and JNK, did not detect the phosphorylated forms of either protein (Figure 6), suggesting that the activation of MAPK and modulation of effects on A549 cell viability or survival might not be through pathways incorporating Src or JNK signaling.

Cisplatin and synthetic curcumin analogs could act on pathways that integrate ROS signaling to kill cancer cells. Cisplatin can induce toxic ROS production in cancer cells [5,40], while curcumin, by either increasing or decreasing ROS levels, can cause A549 cell toxicity [41,42]. EF24 can cause cell death in A549 cells, and structurally similar compounds can kill A549 cells through increased ROS production [10,43]. CLEFMA can increase cell death in the lung cancer cell line, H-411, by increasing ROS yield [19] and can reduce the viability of A549 cancer cells [22]. Our results showed that cisplatin and the positive control, H_2_O_2_, increased ROS production, while both curcuminoids and the combination samples had reduced ROS yields (Figure 2). The reduced ROS levels in curcuminoid–treated samples could be from acting as ROS scavengers like curcumin [44,45]. When cisplatin and curcumin are combined, curcumin ROS scavenging can counteract cisplatin’s induction of ROS [12,13,46]. Thus, CLEFMA and EF24 might be able to scavenge the ROS generated from cisplatin treatment and compensate against this effect to reduce the overall ROS yield. It is possible that the lower ROS levels in the curcuminoid and cisplatin–curcuminoid-treated samples were elevated and could modulate cancer cell toxicity pathways at an earlier time point than the 48-h interval when our ROS analysis was performed. Further, CLEFMA and EF24, like curcumin, may activate cellular mechanisms in cancer cells that counteract and reduce the increased ROS levels [47,48].

Cisplatin, CLEFMA, and EF24 could also signal through different pathways or components of the same pathway to modulate ROS production and cell death. EF24 can promote caspase-3 signaling in ovarian carcinoma cells treated with cisplatin and reduce ROS production [23]. Our caspase-3/7 experiment suggests that cisplatin and both curcuminoids reduced caspase activity, but the combination treatments restored activity to normal (Figure 3C). This result could mean that CLEFMA and EF24 might potentiate cisplatin-mediated caspase-3/7 signaling while decreasing ROS levels. EF24 can form an adduct with the antioxidant glutathione in leukemia cancer cells, causing increased ROS expression [24]. Cisplatin does not target glutathione, but can reduce the activity of glutathione S-transferase, which catalyzes the binding of glutathione to ROS [49]. Although we found that cisplatin treatment increased ROS production in A549 cells (Figure 2), it is possible that either curcuminoid could act on different pathways or pathway components to prevent the platinum compound from increasing ROS levels. Interestingly, the phytochemical ent-11α-hydroxy-15-oxo-kaur-16-en-19-oic-acid, in combination with cisplatin, can induce AIF-modulated apoptosis in A549 cells while reducing ROS production [50], which suggests that AIF may be able to promote cancer cell death without acting through ROS production to activate MAPK or JNK apoptotic signaling. Caspase-12 signaling requires ROS activation [51], and as elevated ROS levels could occur earlier than the 48 h interval that we assayed, it is possible that caspase-12 signaling was activated and modulated MAPK at an earlier time point.

Combining chemotherapy drugs that act on different cell pathways could increase their individual effects against cell migration. Cisplatin can reduce A549 cell migration by acting through multiple pathways including those incorporating transforming growth factor β1 (TGF-β), Sox2, Wnt/β catenin signaling and the microRNA, miR-146a, through regulation of cyclin J [52,53,54]. However, the effect of cisplatin treatment against A549 migration can be mitigated by the development of resistance [52,55]. Curcumin also reduces A549 cell migration by modulating the microRNA, miR-330-5p, and pathways incorporating c-Met, Akt, mTOR, S6, MAPK, TGF-β, and Wnt, and by downregulating cyclin D1 and upregulating p21 mRNA expression [14,56,57]. Curcumin and cisplatin combinations can reduce the invasive property of A549 cells by acting on pathways integrating Bcl-2, Bax, cyclin D1, and p21 signaling [14,58]. Further, the curcuminoid H-4073 can enhance cisplatin’s effect against head and neck cancer migration by inhibiting the STAT, FAK, Akt, and VEGF pathways [59].

As CLEFMA and EF24 are structural analogs of curcumin [19,24], and both cisplatin and curcumin target common and separate cellular migration pathways, we reasoned that these curcuminoids might, alone or in combination with cisplatin, have an effect against cancer cell migration. We found that cisplatin, CLEFMA, or EF24 treatment alone did not reduce A549 migration, but that the combination of cisplatin with either curcuminoid significantly decreased cell migration (Figure 4 and Figure 5). This result suggests that cisplatin and either curcuminoid might act together to potentiate an effect on the same pathway. However, it is also possible that cisplatin and either curcuminoid can act on separate pathways, which could prevent the action of one drug to reduce migration from being compensated against in an alternate pathway. TGF-β and Wnt signaling are targeted by cisplatin and curcumin in migration [14,52,53,54,57], and both pathways can be altered along with MAPK signaling during metastasis [60,61]. Our western blot data (Figure 6), which showed the expression of pMAPK at a 48-h time point, could mean that MAPK signaling was active at the earlier time point than we measured in our migration experiments, and that this protein was involved in suppressing A549 cancer cell migration under conditions of cisplatin and curcuminoid combination treatment (Figure 4 and Figure 5).

Cisplatin treatment damages DNA in auditory hair cells causing ROS generation, hair cell death, and reduced hearing [9,62,63,64,65]. Curcumin treatment can induce antioxidant enzymes and counteract auditory threshold shifting caused by cisplatin [13,66]. Although curcumin can act to scavenge free radicals in some cancer and non-cancer cell types [67,68], its mechanism of action to prevent cisplatin-mediated ototoxicity is not well-understood. EF24 can increase and decrease ROS levels in different cancer cell lines [18,23,24]. CLEFMA has been shown to increase ROS production in cancer cells without increasing ROS levels in non-cancerous cells [19,20]. As our cancer cell ROS data suggested that both curcuminoids could reduce ROS yield and possibly counteract the effect of cisplatin on ROS generation, we used the auditory evoked potential test in a zebrafish model to test if CLEFMA or EF24 could reverse threshold shifts caused by cisplatin treatment.

We found that cisplatin caused significantly increased hearing threshold shifts (up to 10 dB) at three frequencies compared to both the platinum compound’s NaCl vehicle and the curcuminoid vehicle, DMSO, while both CLEFMA- and EF24-treated fish exhibited less than 5-dB threshold shifts at one frequency (CLEFMA) and two frequencies (EF24; Figure 7A,B; Table 2 and Table 3; Appendix A). When we initially treated fish with cisplatin and then with either curcuminoid, we found that both CLEFMA and EF24 significantly reduced threshold shifts at multiple frequencies, as did fish treated with cisplatin followed 24 h later by DMSO vehicle (Figure 7C; Table 2 and Table 3; Appendix A). Analysis of the mean temporary threshold shift data from these experiments suggest that cisplatin with EF24 does not produce an otoprotective effect, but that cisplatin with either CLEFMA or DMSO may do so (Figure 7D; Table 2 and Table 3; Appendix A). However, our results could be interpreted to suggest that DMSO, and not a curcuminoid, could be responsible for any otoprotective benefit. As DMSO can neutralize cisplatin [69,70], it was injected with or without a curcuminoid 24 h after initial cisplatin treatment. This protocol was based on results taken from biochemical assays performed in cultured cancer cells where significant platinum uptake into the nucleus occurs within 3 h of cisplatin treatment [26,71]. However, auditory tissue may exhibit different and slower platinum uptake physiology than in cell culture, resulting in DMSO being able to react with and inactivate significant amounts of cisplatin. DMSO can act as either an antioxidant or pro-oxidant [72]. As an antioxidant, it could target and neutralize ROS generated downstream from cisplatin treatment, causing an otoprotective effect. However, our data (Figure 7) does not suggest that DMSO acts as a pro-oxidant, where increased ROS generation would be expected to damage sensory hair cells resulting in threshold shifts.

## 4. Materials and Methods

### 4.1. Cell Culture

The non-small cell lung cancer cell line, A549, was obtained from American Type Culture Collection (ATCC; Manassas, VA, USA). F12K media (Gibco, Gaithersburg, MD, USA) and was used with 10% fetal bovine serum (FBS; Mediatech, Tewksbury, MA, USA) and 1% penicillin/streptomycin (Gibco). Cells were incubated at 37 °C in 5% CO_2_ with passaging every 4–7 days.

### 4.2. Cellular Viability Assay

The colorimetric 3-(4,5-dimethylthiazol-2-yl)-2,5-diphenyltetrazolium bromide (MTT; Sigma-Aldrich, Milwaukee, WI, USA) assay was used to determine the effect of cisplatin, EF24, and CLEFMA (Sigma-Aldrich) against cancer cell viability. A549 cells were seeded at a density of 5000 cells per well in supplemented F12K media in 96-well plates in replicates of three in three separate experiments and incubated for 24 h at 37 °C in 5% CO_2_. Then, the 96-well plates were treated with a dilution series (500, 50, 5, 0.5, or 0.05 µM) of either cisplatin, EF24, or CLEFMA at 37 °C in 5% CO_2_ over the following time intervals: Cisplatin (0–24 or 0–48 h), EF24 (0–24 or 24–48 h), and CLEFMA (0–24 or 24–48 h). A negative control (cells in media with no drug administration), positive control (cells in media with Triton X-100; Fisher Scientific, Waltham, MA, USA), and media-only blanks were run. After 24 or 48 h, the MTT assay was run for 2 h and absorbance was determined using a spectrophotometer (BioTek, Winooski, VT, USA) set at 570 nm and 690 nm. Then, for experiments where cisplatin and a curcuminoid were combined, another set of plates was prepared using the same procedure as above, except the experimental wells were treated (t = 0 h), with the 48-h IC_50_ value of cisplatin followed by treatment (t = 24 h) with a dilution series (500, 50, 5, 0.5, or 0.05 µM) of a curcuminoid for 24 h. Then, the MTT assay was performed to generate a combination treatment IC_50_ value based on the total concentration of cisplatin with EF24 or CLEFMA. A separate plate with a DMSO (Fisher Scientific)-only treatment was prepared using the same dilution series and the protocol described above to determine whether this solvent used in the curcuminoid preparations affected cellular viability.

### 4.3. Reactive Oxygen Species Assay

A spectrophotometric fluorescent assay adapted from [23,73] was used to measure ROS in cancer cells treated with either cisplatin, a curcuminoid, cisplatin combined with a curcuminoid, a positive control, or a negative control. Seven 10-cm dishes were prepared each with 1 × 10^6^ A549 cells in 10 mL of F12K media with 10% fetal bovine serum and 1% penicillin/streptomycin supplementation. Dishes were incubated for 24 h at 37 °C and 5% CO_2_. Media was then aspirated out of the dishes and replaced with 10 mL of the following treatments (all in F12K media): Media only for 48 h (negative control); 100 µM 30% hydrogen peroxide (H_2_O_2_) (Sigma-Aldrich) for 48 h (positive control); 10 µM cisplatin for 48 h, 2 µM EF24 for 48 h, 15 µM CLEFMA for 48 h, 10 µM cisplatin for 24 h followed by 10 mL of 2 µM EF24 for another 24 h, and 10 µM cisplatin for 24 h followed by 10 mL of 15 µM CLEFMA for another 24 h. After 48 h, media was aspirated out. Each dish was then washed three times with 2 mL of phosphate-buffered saline (PBS) (Sigma-Aldrich). The cells were then detached by adding 1 mL of Accutase (Millipore Sigma, St. Louis, MO, USA) to each dish. Then, the cells were transferred to individual microcentrifuge tubes and 0.5 mL of PBS was added to each tube. Each tube was spun for 5 min at 1000 rpm. The supernatant was then carefully discarded and the pellet was resuspended with 500 µL of 10 µM ROS indicator dye (CM-H_2_DCFDA, Invitrogen, Eugene, OR, USA) in PBS and each tube was incubated for 45 min at 37 °C in 5% CO_2_. Next, the tubes were centrifuged at 1000 rpm for 5 min. Then, the cells were washed three times with PBS and were resuspended in 1 mL of PBS. Then, 100 µL of cell suspension, containing approximately 100,000 cells, was placed into each of nine wells of a black 96-well plate. Also, 100 µL of PBS was placed into nine wells (blank treatment). Then, the plate was placed into a spectrophotometer and read at 495 nm (excitation) and 527 nm (emission) wavelengths.

### 4.4. Caspase Luminescence Assays

Luminogenic kits (Promega, Fitchburg, WI, USA) were used to measure caspase-3/7, -8, and -9 activity. First, 10,000 A549 cells were plated in replicates of three in white 96-well plates (Fisher Scientific) and placed in an incubator for 24 h at 37 °C in 5% CO_2_. Wells were treated with one of the following: cisplatin IC_50_ for 48 h, EF24 IC_50_ from 24–48 h, CLEFMA IC_50_ from 24–48 h, cisplatin IC_50_ for 24 h followed by EF24 IC_50_ for 24 h, cisplatin IC_50_ for 24 h followed by CLEFMA IC_50_ for 24 h, negative control (media with cells), positive control (100 µM 30% hydrogen peroxide (H_2_O_2_), and blank (media with no cells). After introduction of the luminogenic substrate, plates were kept in the dark at room temperature. The plates were read at 0.5, 1, 2, and 3 h-timed intervals with a plate reader (BioTek) using the luminescent setting.

### 4.5. Western Blot Assay

Western blot analysis was used to measure AIF or caspase-12 expression. First, seven 10-cm dishes were seeded with 1.0 × 10^6^ A549 cells and incubated at 37 °C in 5% CO_2_ for 24 h. Next, the media was aspirated out and each dish was treated with one of the following preparations: 10 mL media only (negative control) for 48 h, 10 mL of 100 µM 30% H_2_O_2_ (positive control) for 48 h, 10 mL of IC_50_ cisplatin for 48 h, 10 mL of IC_50_ EF24 from 24–48 h, 10 mL of IC_50_ CLEFMA from 24–48 h, 10 mL of IC_50_ cisplatin for 24 h followed by IC_50_ EF24 for 24 h, or 10 mL of IC_50_ cisplatin for 24 h followed by IC_50_ CLEFMA for 24 h.

Cells were washed with PBS and then removed using a scraper and transferred into micro-centrifuge tubes. After centrifugation (1500 RPM for 5 min), cells were lysed using sodium dodecyl sulfate buffer (SDS, Cell Signaling Technology, Danvers, MA, USA) and then protease and phosphatase inhibitor (Cell Signaling Technology) was added to each tube. Samples were then heated to 95–100 °C followed by sonication and centrifugation (10 min, 14,000× *g*, 4 °C). Determination of protein concentration was done using the Bradford assay (Thermo Scientific, Rockford, IL, USA) and a standard curve of results made using Prism (GraphPad, version 6, La Jolla, CA, USA). Samples containing 15 µg of protein (5 µg for EF24 samples) were then loaded into Mini Protean TGX (Bio-Rad, Hercules, CA, USA) gel wells. Gels were run in a gel apparatus (Bio-Rad) for approximately 1 h, and samples were then electro-transferred for 60 min to a polyvinylidene fluoride (PVDF) membrane (Bio-Rad) using a Bio-Rad transfer device. Membranes were then incubated with primary antibody obtained from Cell Signaling Technology (Danvers, antibody product numbers are in the following parentheses) against AIF (4642), β-tubulin (2146), caspase-12 (2202), GAPDH (3683), JNK (9252), pJNK (4668), MAPK (4370), pMAPK (4695), Src (2109) or pSrc (2101) diluted 1:1000 overnight. After washing in tris-buffered saline (TBS) and polysorbate 20 (TBST, Cell Signaling Technology), membranes were incubated in 10 mL SignalFire™ (Cell Signaling Technology) and were then fluorescence imaged using an Alpha-Innotech FluorChem HD2 Gel Imaging System (San Leandro, CA, USA).

### 4.6. Cell Migration Assay

A cell migration assay was used to investigate how the curcuminoids, alone or in combination with cisplatin, affected cancer cell migration. The A549 cell line was grown as previously described in 10-cm diameter dishes to approximately 90% confluence, and the media was then aspirated out. Then, a wound was made with a pipet tip through the cell monolayer. The cells were then immediately treated with one of the following for 24 h: 15 μM CLEFMA only, 2 μM EF24 only, 10 μM cisplatin only, 10 μM cisplatin with 15 μM CLEFMA, 10 μM cisplatin with 2 μM EF24, or media only (negative control). For the cisplatin with curcuminoid treatments, the curcuminoid was added three hours after the cisplatin. This step was to ensure that the cisplatin can enter the cell and the nucleus [26,71] before the curcuminoid vehicle, DMSO, which can inactivate cisplatin [69,70], was introduced. For treatments where cisplatin and a curcuminoid were combined, the wound was not made until after curcuminoid addition. Each treatment category was performed in triplicate.

A Nikon camera (DS-5M, Tokyo, Japan) attached to a Nikon TMS microscope was used to take pictures of the wound at 0 h and 24 h after treatment. Then, the area of the wound was measured using Adobe Photoshop’s (San Jose, CA, USA) lasso function. The percent wound recovery was then calculated by subtracting the cleared area at 24 h from the cleared area at 0 h and then dividing the resultant value by the original cleared area and converting to a percentage.

### 4.7. Zebrafish Maintenance

Wild-type zebrafish (*Danio rerio*) were obtained from a commercial supplier (Segrest Farms, Gibsonton, FL, USA) and maintained in the Western Kentucky University animal handling facility following Institutional Animal Care and Use Committee protocols (Animal Welfare Assurance #A3558-01). All zebrafish were a mix of male and female adult animals at least six months of age and were maintained according to standard methods [74].

### 4.8. Auditory Evoked Potential

The auditory evoked potential (AEP) technique [75] was performed in zebrafish to assess whether the curcuminoids counteracted cisplatin-induced hearing threshold shifts. Zebrafish were microinjected with cisplatin, a curcuminoid, cisplatin with a curcuminoid, cisplatin vehicle (0.9% sodium chloride) (Sigma-Aldrich), or curcuminoid vehicle (DMSO). Cisplatin-injected animals were injected with 25 mg cisplatin/kg body weight. Curcuminoid-treated fish were given 5 mg compound/kg body weight injections. Vehicle-treated fish were injected with volumes equivalent to their experimental counterparts by body weight. Fish injected with only cisplatin, EF24, CLEFMA, or vehicle were subjected to AEP analysis 48 h after injection. However, fish treated with a combination of cisplatin and a curcuminoid were injected with cisplatin at time 0 h, injected with a curcuminoid at time 24 h, and then analyzed with the AEP technique at 48 h.

Fish were prepared for AEP by being anaesthetized with tricaine methanesulfonate (MS-222) (Argent, Redmond, WA, USA) and then placed into a mesh harness suspended 6 cm from the water surface and 22 cm above a University Sound UW-30 underwater speaker (Electro-Voice, Burnsville, MN, USA) in a 19-L tank containing 27–28 °C water. Electrical interference was minimized by keeping the tank within a Faraday cage inside a sound-attenuation room (WhisperRoom, Inc., Knoxville, TN, USA). Three stainless steel subdermal electrodes (27 gauge; Rochester Electro-Medical, Inc., Tampa, FL, USA) were attached 1–2 mm subdermally into the fish at the following locations: over the brainstem (recording electrode), between the nares (reference electrode), and in the tail musculature (ground electrode). Pure tone pip sound stimuli at eight different frequencies (100, 250, 400, 600, 800, 1000, 1500, and 3000 Hz) were presented to the fish and AEP waveforms collected using SigGen and BioSig software running on a TDT physiology system (Tucker Davis Technologies, Inc., Alachua, FL, USA). The sound pressure levels of each frequency were confirmed using a calibrated hydrophone (GRAS Type 10CT, Denmark) placed proximate to the fish. Each frequency was tested by decreasing decibel levels in 5-dB steps until an AEP trace was no longer visible. The last sound pressure level at which an AEP trace was visible was noted as the threshold for each frequency, and the frequency thresholds were used to produce audiograms. Between six and eight fish were treated for each treatment category.

### 4.9. Statistical Analysis

GraphPad Prism v6 (La Jolla, CA, USA) with the *p* value set at 0.05 was used for all statistical analysis unless otherwise noted below. MTT assay IC_50_ values were calculated in Prism using a relative, nonlinear best fit analysis performed with the sigmoidal, 4PL, x is log(concentration) analysis feature or were calculated using a linear analysis with ED50plus v1.0 online software (Mexico City, Mexico). MTT assay standard deviation values were calculated using ED50plus. The second-order polynomial (quadratic) function in Prism was used to derive best fit values and linear regression analysis was used to analyze slope difference. AEP, caspase, and ROS assay results were analyzed using a two-way ANOVA and Tukey’s post hoc tests. Migration assay results were analyzed using a one-way ANOVA with a Dunnett’s multiple comparison test. Analysis of cisplatin-curcuminoid drug combinations to determine additive or synergistic effects was performed using the Prism Bliss independence nonlinear analysis feature with the confidence limit set to 95%.

## 5. Conclusions

Cisplatin is extensively used in chemotherapy but is limited by the development of resistance and side effects, including permanent hearing loss. The synthetic curcumin analogs, CLEFMA and EF24, may signal in different cell death pathways to potentiate the effect of cisplatin against cancer and prevent its side effects. In this project, we found that cisplatin and the curcuminoids, alone or in combination, had similar effects against A549 non-small lung cancer cell viability. Also, we found that cisplatin treatment acted to increase ROS production, but that the curcuminoids, separately or in combination, reduced ROS yield. Further, the treatments either reduced or left the activity of caspases-3/7, -8, and -9 unchanged. Western blot analysis suggested that the compounds may modulate cell death or survival mechanisms integrating AIF, caspase-12, and MAPK signaling, but not JNK or Src. We also found that the cisplatin–curcuminoid combinations could reduce A549 cancer cell migration. Finally, our AEP analysis suggests that the curcuminoids may be able to reduce cisplatin-induced auditory threshold shifts, but we cannot rule out a role for the curcuminoid vehicle, DMSO, in this effect. Evidently, additional study of cisplatin, CLEFMA, and EF24 could help determine their mechanistic relationships in modulating cancer cell death and ototoxicity and assist in the development of new chemotherapy drugs with reduced side effects.

## Figures and Tables

**Figure 1 molecules-24-03889-f001:**
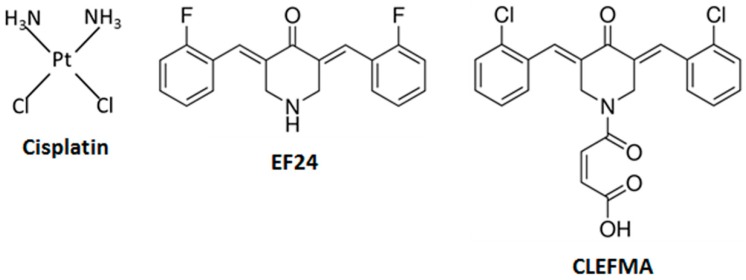
Chemical structures of cisplatin, and the curcuminoids, CLEFMA and EF24, investigated in this project. Cisplatin is a bifunctional platinum(II) complex with two chloride leaving ligands. EF24 and CLEFMA are diphenyldihaloketone analogs with either fluorine (EF24) or chlorine substituents (CLEFMA) (CLEFMA and EF24 structures are modified from sigma-aldrich.com).

**Figure 2 molecules-24-03889-f002:**
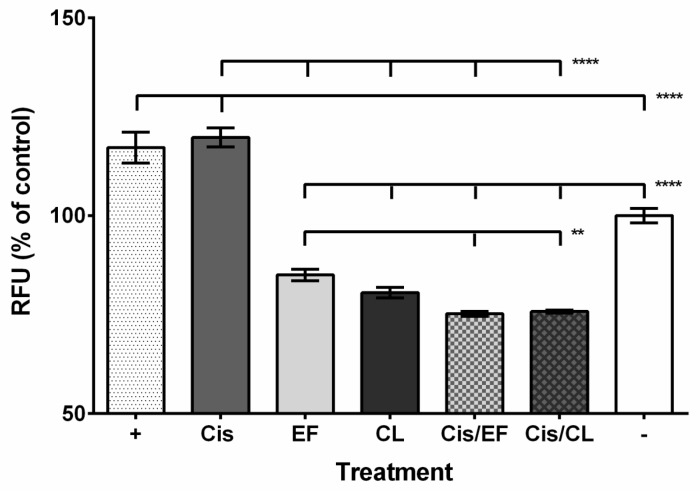
Effect of cisplatin, curcuminoid, and cisplatin–curcuminoid combination treatments on ROS production in A549 cells. Positive control (H_2_O_2_) and cisplatin treatment causes increased ROS production, whereas curcuminoid alone and curcuminoid with cisplatin treatment caused decreased ROS yield in A549 cells compared to control. Abbreviation key: “+” = positive control (H_2_O_2_); “Cis” = cisplatin; “EF” = EF24; “CL” = CLEFMA; “Cis/EF” = cisplatin with EF24; “Cis/CL” = cisplatin with CLEFMA; “−“ = negative control. *n* = 9; “**”, *p* < 0.01; “***”, *p* < 0.001.

**Figure 3 molecules-24-03889-f003:**
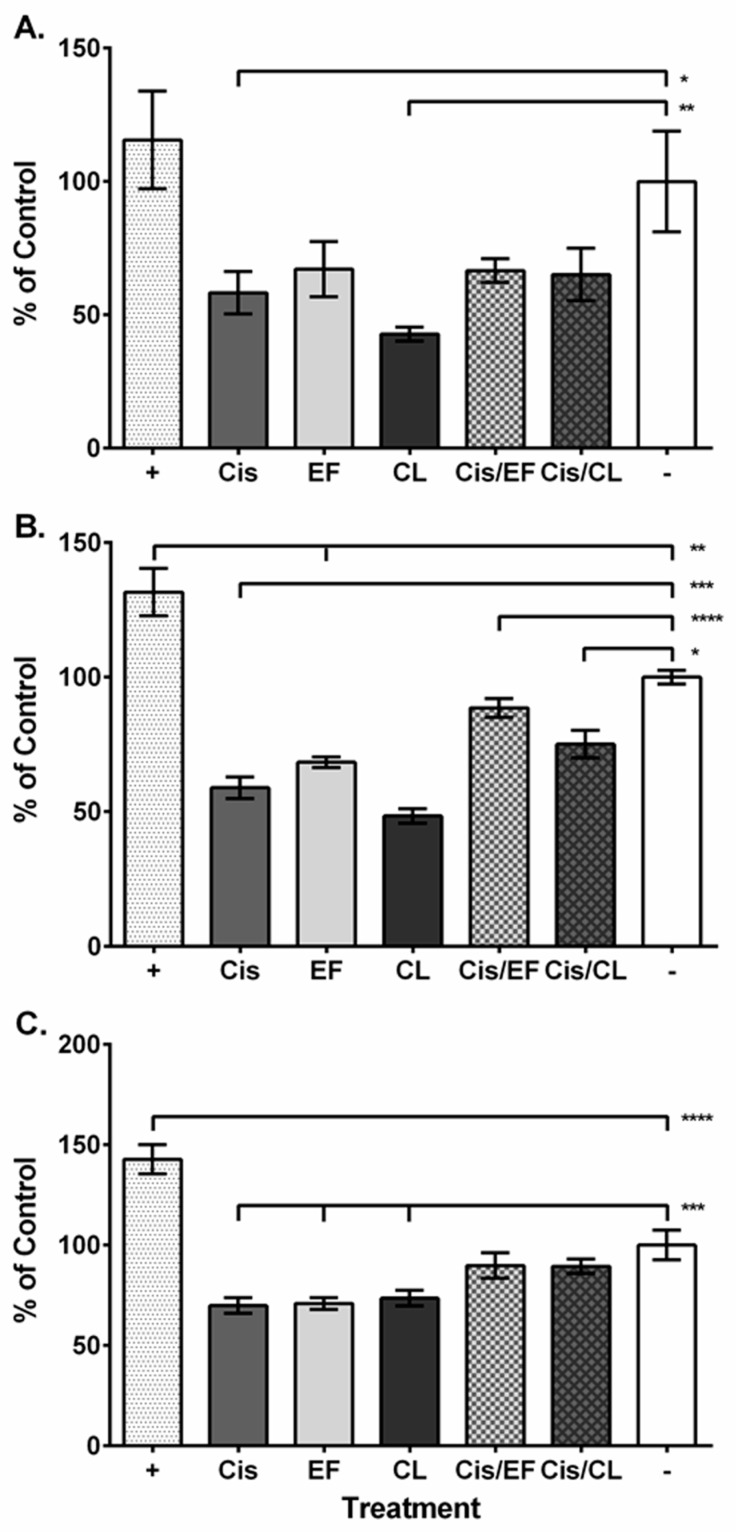
Effect of cisplatin, curcuminoid, and combination treatments on caspase activity in A549 cells. (**A**–**C**) Labeling: Positive control (“+”; white with gray speckled pattern), negative control (“−”; white), cisplatin (“Cis”; gray), EF24 (“EF”; light gray), CLEFMA (“CL”; dark gray), cisplatin and EF24 (“Cis/EF”; gray and light gray with box pattern), cisplatin and CLEFMA (“Cis/CL”; gray and dark gray with diamond pattern). (**A**) At 2 h from assay initiation, caspase-8 activity decreased for cisplatin and CLEFMA treatments. (**B**) At the same time point, caspase-9 activity decreased for all treatment categories except for cisplatin with EF24 and the positive control, which had increased activity. (**C**) Caspase-3,-7 activity decreased for cisplatin and both curcuminoids 2 h from assay initiation, whereas the positive control caused increased activity. *n* = 3; “*”, *p* < 0.05; “**”, *p* < 0.01; “***”, *p* < 0.001; “****”, *p* < 0.0001.

**Figure 4 molecules-24-03889-f004:**
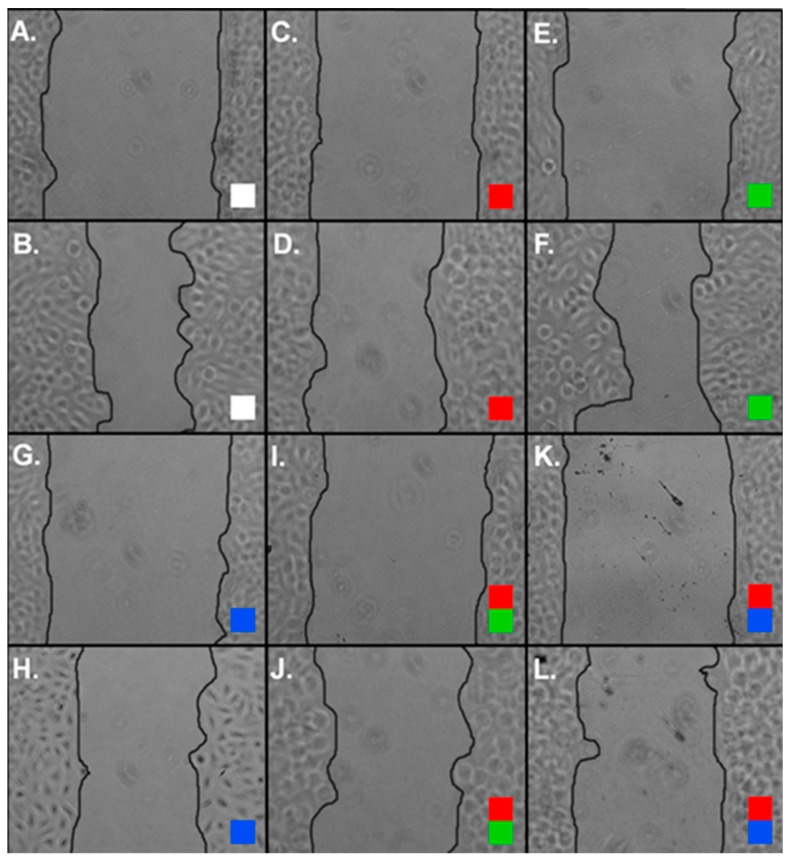
Representative images of A549 wound recovery after vehicle and experimental treatments. (**A**–**L**) Color code (white box = control; red box = cisplatin only; green box = EF24 only; blue box = CLEFMA only; red and green box = cisplatin with EF24; red and blue box = cisplatin with CLEFMA. (**A**) Control (0 h). (**B**) Control (24 h). (**C**) Cisplatin (0 h). (**D**) Cisplatin (24 h). (**E**) EF24 (0 h). (**F**) EF24 (24 h). (**G**) CLEFMA (0 h). (**H**) CLEFMA (24 h). (**I**) Cisplatin–EF24 (0 h). (**J**) Cisplatin–EF24 (24 h). (**K**) Cisplatin–CLEFMA (0 h). (**L**) Cisplatin–CLEFMA (24 h).

**Figure 5 molecules-24-03889-f005:**
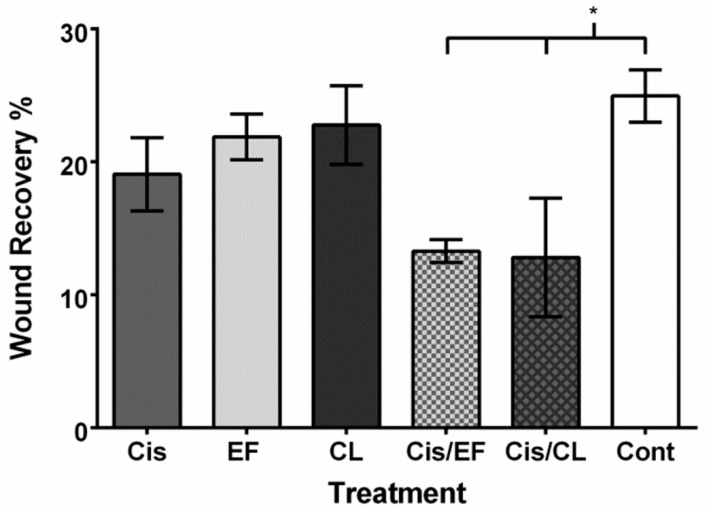
Cisplatin treatment combined with either CLEFMA or EF24 reduced A549 wound recovery. Abbreviation key: “Cis” = cisplatin; “EF” = EF24; “CL” = CLEFMA; “Cis/EF” = cisplatin with EF24; “Cis/CL” = cisplatin with CLEFMA; “Cont“ = negative vehicle control. Treatment with cisplatin, EF24, and CLEFMA caused nonsignificant decreases in cancer cell wound recovery. However, the combination of cisplatin with either EF24 or CLEFMA caused significantly decreased migration into areas cleared of A549 cells. *n* = 3; *p* < 0.05.

**Figure 6 molecules-24-03889-f006:**
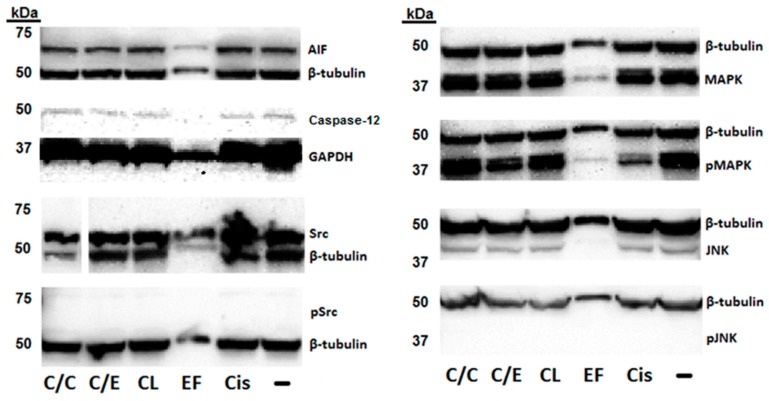
Effect of cisplatin and two curcuminoids on proteins modulating cell death. Labeling: Negative control (“-“), cisplatin (“Cis), EF24 (“EF”), CLEFMA (“CL”), cisplatin and EF24 (“C/E”), cisplatin and CLEFMA (“C/C”). Numbers on left of blots represent molecular weight in kilodaltons (kDa). Western blot results show detection of apoptosis-inducing factor (AIF), caspase-12 (except EF24), c-Jun N-terminal kinases (JNK), mitogen-activated protein kinase (MAPK), phosphorylated MAPK (pMAPK), proto-oncogene tyrosine-protein kinase (Src), and the controls, β-tubulin and GAPDH, throughout all treatment categories. Both pJNK and phosphorylated Src (pSrc) proteins were not detected for all treatment categories.

**Figure 7 molecules-24-03889-f007:**
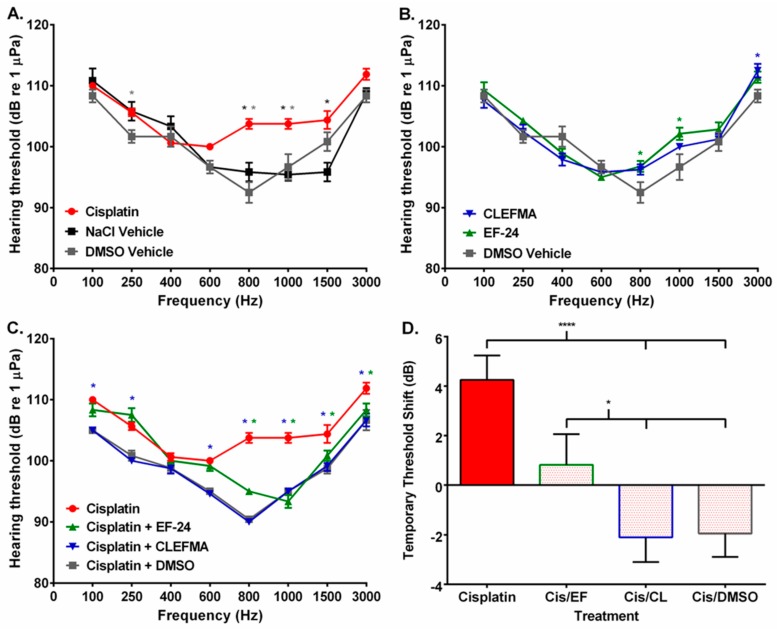
Curcuminoid treatment reduced auditory threshold shifts in cisplatin-treated zebrafish. (**A**) Cisplatin- (red line) treated fish had increased auditory thresholds at several frequencies compared to cisplatin vehicle- (NaCl; black line) and curcuminoid vehicle- (DMSO; gray line) treated fish. (**B**) CLEFMA- (blue line) and EF24- (green line) treated fish had auditory thresholds similar to curcuminoid DMSO vehicle. (**C**) Administration of CLEFMA, EF24, and DMSO 24 h after cisplatin treatment reduced fish hearing thresholds compared to cisplatin treatment alone. Line key: Cisplatin (red line); cisplatin–EF24 (green line); cisplatin–CLEFMA (blue line); cisplatin–DMSO (gray line). (**D**) Mean ± SE temporary threshold shifts of zebrafish 48 h after cisplatin, cisplatin–EF24, cisplatin–CLEFMA, or cisplatin–DMSO treatment (Key: “Cis/EF” = cisplatin with EF24; “Cis/CL” = cisplatin with CLEFMA; “Cis/DMSO” = cisplatin with DMSO). *n* = 6–8; “*”, *p* ≤ 0.05; “****”, *p* ≤ 0.001. Colored asterisks refer to the color of the curcuminoid, vehicle, or cisplatin–curcuminoid/vehicle combination being compared to cisplatin. See Table 2 and Table 3 and Appendix A for ANOVA analysis results.

**Table 1 molecules-24-03889-t001:** IC_50_ values for cisplatin, curcuminoid, and combination treatments in A549 cell culture. Standard deviation values are provided after the ± symbol for each inhibitory concentration value. Cell culture treatment times are indicated.

Treatment	IC_50_ (µM)
Cisplatin (24 h)	10.91 ± 0.19
Cisplatin (48 h)	7.49 ± 0.16
CLEFMA (0–24 h)	13.82 ± 0.18
CLEFMA (24–48 h)	16.05 ± 0.15
EF24 (0–24 h)	1.74 ± 0.28
EF24 (24–48 h)	2.47 ± 0.14
Cisplatin (48 h) + CLEFMA (24–48 h)	2.19 ± 0.17
Cisplatin (48 h) + EF24 (24–48 h)	2.94 ± 0.09

**Table 2 molecules-24-03889-t002:** Statistical results for auditory evoked potential testing comparisons between experimental treatments and cisplatin or cisplatin vehicle. Hearing thresholds from cisplatin, EF24, CLEFMA, cisplatin–EF24, or cisplatin–CLEFMA combination-treated zebrafish were compared with cisplatin or cisplatin vehicle- (NaCl) treated fish. Key: “Cisplatin/EF24” = cisplatin followed by EF24, “Cisplatin/CLEFMA” = cisplatin followed by CLEFMA. “ns” = nonsignificant, *p* > 0.05; “*”, *p* < 0.05; “**”, *p* < 0.01; “****”, *p* < 0.0001.

Freq (Hz)	Cisplatin	EF24		CLEFMA		Cisplatin/EF24		Cisplatin/CLEFMA	
	Vehicle	Vehicle	Cisplatin	Vehicle	Cisplatin	Vehicle	Cisplatin	Vehicle	Cisplatin
100	ns	ns	ns	ns	ns	ns	ns	**	**
250	ns	ns	ns	ns	ns	ns	ns	**	**
400	ns	*	ns	**	ns	ns	ns	*	ns
600	ns	ns	**	ns	*	ns	ns	ns	**
800	****	ns	****	ns	****	ns	****	**	****
1000	****	****	ns	*	ns	ns	****	ns	****
1500	****	****	ns	**	ns	*	ns	**	ns
3000	ns	ns	ns	ns	ns	ns	ns	ns	**

**Table 3 molecules-24-03889-t003:** Statistical results for auditory evoked potential testing comparisons between experimental treatments and the curcuminoid vehicle (DMSO). Hearing thresholds from cisplatin, EF24, CLEFMA, cisplatin–EF24, or cisplatin–CLEFMA combination-treated zebrafish were compared with curcuminoid vehicle- (DMSO) treated fish. Key: “Cisplatin/EF24” = cisplatin followed by EF24, “Cisplatin/CLEFMA” = cisplatin followed by CLEFMA, “Cisplatin/DMSO” = cisplatin followed by DMSO. “ns” = nonsignificant, *p* > 0.05; “*”, *p* < 0.05; “**”, *p* < 0.01; “****”, *p* < 0.0001.

Freq (Hz)	Cisplatin	EF24	CLEFMA	Cisplatin/EF24	Cisplatin/CLEFMA	Cisplatin/DMSO	
	DMSO	DMSO	DMSO	DMSO	DMSO	DMSO	Cisplatin
100	ns	ns	ns	ns	ns	ns	**
250	ns	ns	ns	**	ns	ns	*
400	ns	ns	ns	ns	ns	ns	ns
600	ns	ns	ns	ns	ns	ns	**
800	****	*	ns	ns	ns	ns	****
1000	****	**	ns	ns	ns	ns	****
1500	ns	ns	ns	ns	ns	ns	**
3000	ns	ns	ns	ns	ns	ns	**

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
