# Peer review of "Anti-Cancer and Ototoxicity Characteristics of the Curcuminoids, CLEFMA and EF24, in Combination with Cisplatin"

_molecules, 2019, doi:10.3390/molecules24213889_

Round 1
Reviewer 1 Report
Thank you for addressing my comments and concerns.
Author Response
Reviewer 1 did not recommend any additional changes. We thank the reviewer for his/her helpful feedback.
Reviewer 2 Report
This paper presents In Vitro combination treatments of curcuminoids with cisplatin. This paper is interesting, I recommend publication after conducting the following minor revisions:
The experimental section is lacking description of the source of compounds, and please clarify better the sentence: "to generate a combination treatment IC50 value"; if this means that the combination IC50 is based on total concentration of active ingredient cisplatin+curcumin derivative (as can be deduced from the plots in the supplementary material) - please specify clearly.
Author Response
Reviewer 2:
I recommend publication after conducting the following minor revisions:
1) The experimental section is lacking description of the source of compounds, and
2) please clarify better the sentence: "to generate a combination treatment IC50 value"; if this means that the combination IC50 is based on total concentration of active ingredient cisplatin+curcumin derivative (as can be deduced from the plots in the supplementary material) - please specify clearly.
Action taken:
1) We added the source of the compounds to the manuscript immediately after where they were first mentioned in the Materials and Methods section for the following: MTT, Triton X-100, hydrogen peroxide, phosphate buffered saline, ROS indicator dye (H2DCFDA), SDS, protease and phosphatase inhibitors, Bradford assay kit, TBST, sodium chloride, and methanesulfonate. We believe that all other chemicals have their source of origin indicated immediately after their first reference in the Materials and Methods section.
2) We added the language suggested by the reviewer to clarify the sentence ending on line 381 of the manuscript.
We also corrected a punctuation error on line 496.
We thank Reviewer 2 for his/her thoughtful recommendations.
This manuscript is a resubmission of an earlier submission. The following is a list of the peer review reports and author responses from that submission.
Round 1
Reviewer 1 Report
This study investigates whether curcuminoids can improve cisplatin efficacy and decrease associated side effects. The topic is of significant interest, and appropriate methodologies have been used. A key strength is inclusion of the zebrafish experiments. It would have been helpful to perform some of the studies in additional cell lines; the use of one cell line is a weakness. I have the following specific concerns and suggestions;
1. I recommend removing the opening 2 sentences (lines 12 -16) from the abstract; this information belongs in the introduction section. Instead, start the abstract with the goal of the study, e.g. ‘the goal of this study was to determine whether curcuminoids can improve cisplatin efficacy and reduce cisplatin-related toxicities’.
2. Specific details regarding the impact of curcuminoids should be included in the abstract, e.g. report the fold change in cell viability (and include a p value) etc.
3. I recommend including a conclusion sentence in the abstract.
4. Lines 37-40: please provide a brief description of the mechanism by which curcumin mediates ROS production in cancer cells, and explain why this does not occur in ‘normal’ cells.
5. Lines 63-74: this information should not be included here/should be removed.
6. Line 87 to 88: The sentence states ‘After evaluating the CLEFMA and 88 EF24 IC50 values, we rounded them to 2 and 15 μM respectively for subsequent experiments’ – are these numbers accidently switched?
7. To make it easier for the reader to follow, please include a sentence which compares the IC50 for CLEFMA alone versus cisplatin + CLEFMA and compares the IC50 for E2F4 alone versus cisplatin + E2F4 (it may also be helpful to also state the fold-change for these comparisons). The observation that the cisplatin + E2F4 treatment is similar to treatment with E2F4 alone should be commented on in the discussion section.
8. Table 1: it is confusing to include a range of time for the treatments – instead, simply state the amount of exposure time (e.g. 24 hours).
9. Figure 3: I recommend that the same formatting be used as for figure 2, i.e. the use of bars to show statistical comparisons/significance, and use of ‘-‘ for negative control.
10. Figures 2, 3, 4, and 5; please report fold changes in the results section relating to each of these figures – this allows the reader to get an idea of the relative magnitude of the changes as well as whether or not they are significant.
11. Figure 6, lines 163 - 173: the authors need to comments on the differences that were observed between treatments, not simply state that these molecules were expressed. E.g. It appears that the E2F4 treatment caused decreased expression of these molecules.
12. Figure 6: the labeling for the different blots shown in this figure needs to be fixed – as is, it is very confusing. Perhaps the second panel down on the left can be inverted so that the treatment order is the same as the top left panel?
13. Tables 2 and 3: I suggest including fold changes in these tables as well as statistical significance.
14. Line 245: Please expand on the statement that ‘cisplatin can act through caspase-independent mechanisms…’ to give specific details and relate these to the study findings.
15. Lines 255 to 257: Please explain this statement.
Author Response
Molecules Reviewer Comments
Comments and Suggestions for Authors
This study investigates whether curcuminoids can improve cisplatin efficacy and decrease associated side effects. The topic is of significant interest, and appropriate methodologies have been used. A key strength is inclusion of the zebrafish experiments. It would have been helpful to perform some of the studies in additional cell lines; the use of one cell line is a weakness. I have the following specific concerns and suggestions;
I recommend removing the opening 2 sentences (lines 12 -16) from the abstract; this information belongs in the introduction section. Instead, start the abstract with the goal of the study, e.g. ‘the goal of this study was to determine whether curcuminoids can improve cisplatin efficacy and reduce cisplatin-related toxicities’.We removed the text and rewrote the abstract with a modified version of the suggested opening statement. Specific details regarding the impact of curcuminoids should be included in the abstract, e.g. report the fold change in cell viability (and include a p value) etc.
We introduced considerably more detail to the results section of the abstract, and sought to more precisely present the general results, but due to the 200 word abstract limitation, left more nuanced presentation of the analytical details to the results section of the manuscript. I recommend including a conclusion sentence in the abstract.
We introduced a conclusion as requested. Lines 37-40: please provide a brief description of the mechanism by which curcumin mediates ROS production in cancer cells, and explain why this does not occur in ‘normal’ cells.
We have inserted two new sentences at this point to explain the action of curcumin in ROS production in cancer and non-cancer cells. Lines 63-74: this information should not be included here/should be removed.
We removed the paragraph as suggested. Line 87 to 88: The sentence states ‘After evaluating the CLEFMA and 88 EF24 IC50 values, we rounded them to 2 and 15 μM respectively for subsequent experiments’ – are these numbers accidently switched?
We corrected the switched text as suggested by the reviewer. To make it easier for the reader to follow, please include a sentence which compares the IC50 for CLEFMA alone versus cisplatin + CLEFMA and compares the IC50 for E2F4 alone versus cisplatin + E2F4 (it may also be helpful to also state the fold-change for these comparisons). The observation that the cisplatin + E2F4 treatment is similar to treatment with E2F4 alone should be commented on in the discussion section.
The second reviewer requested that we remove the data information found in the tables from the text (Reviewer 2: minor format-related point #1). In order to reconcile these two reviewer comments, we left the discussion of Table 1 data, which reports the IC50 values, intact, but we removed most of the data information in the text corresponding to Table 2-3 to address the second reviewer’s concern. We combined the two sentences in lines 86 to 88 to more smoothly state what the curcuminoid values are between 24-48 hours. Then, we followed with a sentence stating the combination (cisplatin + curcuminoid) values. We believe that this new set of sentences reasonably provides the information that was requested by this reviewer. Table 1: it is confusing to include a range of time for the treatments – instead, simply state the amount of exposure time (e.g. 24 hours).
We corrected the cisplatin entries in the table to avoid confusion; however, simply stating the exposure time for the curcuminoids would introduce confusion because the time range information would be the same for all curcuminoids in the table, and it would make it impossible to distinguish which curcuminoid IC50 value was applied with cisplatin in the combination experiments. Figure 3: I recommend that the same formatting be used as for figure 2, i.e. the use of bars to show statistical comparisons/significance, and use of ‘-‘ for negative control.
We made the changes requested by the reviewer. Figures 2, 3, 4, and 5; please report fold changes in the results section relating to each of these figures – this allows the reader to get an idea of the relative magnitude of the changes as well as whether or not they are significant.
We removed the percentage values and replaced them with fold change equivalents, except that we retained the percentage values given with figure 5 and added the corresponding fold change values. Figure 6, lines 163 - 173: the authors need to comments on the differences that were observed between treatments, not simply state that these molecules were expressed. E.g. It appears that the E2F4 treatment caused decreased expression of these molecules.
We have added two sentences at the end of the western blot results to describe observed differences between treatments as requested by the reviewer. Figure 6: the labeling for the different blots shown in this figure needs to be fixed – as is, it is very confusing. Perhaps the second panel down on the left can be inverted so that the treatment order is the same as the top left panel?
We performed this correction. Tables 2 and 3: I suggest including fold changes in these tables as well as statistical significance.
We have provided two supplementary tables (Supplementary Table 1 and 2) which provide the requested fold change data using the same tabular format as in Tables 2 and 3. Line 245: Please expand on the statement that ‘cisplatin can act through caspase-independent mechanisms…’ to give specific details and relate these to the study findings.
We added a new sentence and modified the surrounding sentences to expand on this statement and better relate it to the project results. Lines 255 to 257: Please explain this statement.
We have altered this sentence and provided two new following sentences to clarify this statement.

Reviewer 2 Report
This paper by Smith et al describes the effect of combining cisplatin with curcumin synthetic derivatives in various aspects: cytotoxicity, ROS production, protein levels and function, ototoxicity and more. The paper is interesting and describes a relatively broad set of experiments. There are, however, some flaws to my opinion in the measurements’ analysis / description (perhaps also in the experiments themselves – one cannot tell due to missing information). I thus recommend the following revisions:
Major points:
1. There are no cytotoxicity plots, only report of IC50 values. The plots should be provided in a supporting information file (I could not find one; if it was attached and I missed it – my apologies; in fact, my entire review is based on the main manuscript file alone).
2. When analyzing a combination for activity, it should be stated whether the combination gives an antagonistic, an additive or a synergistic behavior. There are some comments in the manuscript on the comparison with no proper analysis (such as “Interestingly, the combination of cisplatin and either curcuminoid caused lower cell viability than cisplatin or CLEFMA treatment alone” line 236). The results should be analyzed based on, for example, isobolography or an equivalent method. Normally a ratio of the two compounds for analysis is determined based on the individual IC50s and then several concentrations of the combination is tested to obtain a meaningful IC50 for the combination in a set ratio.
3. On the same topic: how were the IC50 calculated for the combination? Which concentration was referred to, of one of the drugs? Both (sum of active ingredient)? This is not mentioned. Better care and description of details are required.
4. How were repetitions conducted? No repetitions on separate days? Plates at least? How were the error values determined? And the IC50s, determined how? Manually? Based on a fit? Absolute or relative? All these data are missing.
5. How were the compounds acquired? Re-synthesized? Purchased? From where? This should be mentioned in the methods section.
Minor format-related points:
1. There is no need to write-out information provided in Tables. That’s what table are for, to avoid excessive writing. It is a burden on the reader. The authors should refer the reader to the Tables in the beginning of the relevant paragrapg, and mention only trends, main / average result / boarder points.
2. Table format is different for each Table. Obviously, these would be amended by the editorial if/when accepted, but for scholarly presentation, it is advised to provide consistent format (that of tables 2-2 is normally customary rather than that of Table 1)
3. The size of Figures was not given much thought. Some are huge (2,5), some are small with much space around. No consistency. For example, Figure 3 would better include the three graphs horizontally.
4. Figure 5: please recheck the caption: inconsistency between “-“ and “cont”.
Author Response
Comments and Suggestions for Authors
This paper by Smith et al describes the effect of combining cisplatin with curcumin synthetic derivatives in various aspects: cytotoxicity, ROS production, protein levels and function, ototoxicity and more. The paper is interesting and describes a relatively broad set of experiments. There are, however, some flaws to my opinion in the measurements’ analysis / description (perhaps also in the experiments themselves – one cannot tell due to missing information). I thus recommend the following revisions:
Major points:
There are no cytotoxicity plots, only report of IC50 values. The plots should be provided in a supporting information file (I could not find one; if it was attached and I missed it – my apologies; in fact, my entire review is based on the main manuscript file alone).We have provided the plots as requested in a supplementary attachment. When analyzing a combination for activity, it should be stated whether the combination gives an antagonistic, an additive or a synergistic behavior. There are some comments in the manuscript on the comparison with no proper analysis (such as “Interestingly, the combination of cisplatin and either curcuminoid caused lower cell viability than cisplatin or CLEFMA treatment alone” line 236). The results should be analyzed based on, for example, isobolography or an equivalent method. Normally a ratio of the two compounds for analysis is determined based on the individual IC50s and then several concentrations of the combination is tested to obtain a meaningful IC50 for the combination in a set ratio.
Our graphical software does not provide an option to perform isobolography analysis; however, we have provided two plots in the supplemental materials (Supplemental Figure 3) showing the dose dependent relationships between cisplatin, each curcuminoid and their respective combinations presented as fractional response vs. dosage. Our subsequent analysis of these lines found that both combination treatments did not have significantly different slopes than cisplatin or either curcuminoid. We agree with the reviewer that several statements in the discussion were not clearly stated and have rewritten them to address the new analysis presented in the supplementary materials. The primary purpose of the project was to attempt to elucidate the molecular mechanisms through which cisplatin and the curcuminoids alone or in combination acted, and how they acted against A549 cell migration and as an otoprotectant, but we do not believe that the combination of these compounds causes an additive or synergistic behavior in terms of an effect against A549 cellular viability. On the same topic: how were the IC50 calculated for the combination? Which concentration was referred to, of one of the drugs? Both (sum of active ingredient)? This is not mentioned. Better care and description of details are required.
We reworded the combination experimental procedure given in Methods section 4.4 to clarify that the cells were initially treated with the 48 hour cisplatin IC50 value, and then were treated after 24 hours (to avoid temporally proximate addition to cisplatin of the curcuminoid solvent (DMSO), which chemically inactivates this platinum compound) with a concentration series of either curcuminoid for the purpose of calculating the combination IC50 value using the MTT assay. Therefore, the 48 hour cisplatin IC50 concentration was applied first, at t = 0; then, after 24 hours, a concentration series (500, 50, 5, .5, .05 μM) of either curcuminoid was applied to the wells. Performing the MTT assay on samples prepared in this manner allowed us to derive a combination concentration value. We also noticed that we did not correctly provide some of the timed interval information used in the derivation of the IC50 values, both for cisplatin and the curcuminoids, and we have now added this relevant information to Methods section 4.2. How were repetitions conducted? No repetitions on separate days? Plates at least? How were the error values determined? And the IC50s, determined how? Manually? Based on a fit? Absolute or relative? All these data are missing.
Experimental repetitions were not conducted on separate days, but as replicates within the same experimental plate. In the MTT assay, there were 6 replicates (lines 377-378). In the ROS assay, there were 9 replicates (lines 408-409)[we did correct an error in the replicate N value in the caption for Figure 2]. Caspase assays had 3 replicates (lines 414-415). For the cell migration assay, there were replicates of 3 dishes (line 459). In the zebrafish experiments, 6-8 replicates was performed for each experiment (this information is now added to the manuscript at the end of section 4.8.).
A description of how the error values were generated is provided in section 4.9., Statistical Analysis, where these are listed (We did add the missing information for the caspase assay). In section 4.9., we stated the software used and the type of analysis, e.g., sigmoidal, 4PL, x is log(concentration) analysis or a linear best fit analysis using ED50 software (we corrected the text to include usage of ED50), but we did add that the direct Prism or Prism derived analysis used either a non-linear relative, best fit analysis or a linear analysis based on y=mx + b. We also stated here (lines 502-503) that ED50 software was used to derive standard deviation values for the IC50 experiments.
How were the compounds acquired? Re-synthesized? Purchased? From where? This should be mentioned in the methods section.This information was provided in the methods, section 4.2. (line 379 in the original manuscript), in the description of the cellular viability assay, where it states that the cisplatin, CLEFMA and EF24 compounds were obtained commercially from Sigma-Aldrich, Milwaukee, WI.
Minor format-related points:
There is no need to write-out information provided in Tables. That’s what table are for, to avoid excessive writing. It is a burden on the reader. The authors should refer the reader to the Tables in the beginning of the relevant paragrapg, and mention only trends, main / average result / boarder points.Our interpretation of the other reviewer’s comment #7 is that the IC50 data from Table 1 should not only be retained in the text, but additional data information comparing the curcuminoids with the combination treatments needed to be provided in the text. In order to accommodate the other reviewer’s comment, we have kept the data that is also in Table 1 in the text, but to attempt to address this comment, we have largely removed the data associated with Tables 2 and 3 from the Results section text. Table format is different for each Table. Obviously, these would be amended by the editorial if/when accepted, but for scholarly presentation, it is advised to provide consistent format (that of tables 2-2 is normally customary rather than that of Table 1)
We have removed the left and right borders on Table 1 to make them conform with Tables 2 and 3. The size of Figures was not given much thought. Some are huge (2,5), some are small with much space around. No consistency. For example, Figure 3 would better include the three graphs horizontally.
We have enlarged the smaller figures, including Figure 3, to more completely fill up the spaces provided. However, placing the three panels of Figure 3 horizontally within the margins as suggested would result in the text being too small compared to the other figures. Figure 5: please recheck the caption: inconsistency between “-“ and “cont”.
We have made the requested correction.
Submission Date
17 July 2019
Date of this review
22 Jul 2019 08:40:56

Round 2
Reviewer 2 Report
My recommendation is reject, as some of my main concerns were not addressed: 1. Repeats should be on separate days independently (with plots as viability vs control on the Y axes and not absorbance which means nothing). 2. Isobolographs can be ,mad manually and do not require a particular software.